# Physical Activity, Fitness, and Cognitive Performance of Estonian First-Grade Schoolchildren According Their MVPA Level in Kindergarten: A Longitudinal Study

**DOI:** 10.3390/ijerph18147576

**Published:** 2021-07-16

**Authors:** Kirkke Reisberg, Eva-Maria Riso, Jaak Jürimäe

**Affiliations:** 1Institute of Sports Sciences and Physiotherapy, University of Tartu, 51005 Tartu, Estonia; eva-maria.riso@ut.ee (E.-M.R.); jaak.jurimae@ut.ee (J.J.); 2Tartu Healthcare College, 50411 Tartu, Estonia

**Keywords:** children, physical activity, physical fitness, cognitive skills

## Abstract

Little is known about the longitudinal trajectories and associations regarding physical activity (PA), physical fitness (PF), and cognitive skills in childhood. Accelerometer-based PA, sedentary behavior (SB), PF, and cognitive skills were measured in Estonian children (*n* = 147) in kindergarten (6.6 years) and again at school (7.6 years). Children were subgrouped into lower and upper quartiles by their moderate-to-vigorous PA (MVPA) at 6.6 years. Children in the upper quartile had lower SB, higher PA, and greater muscular strength. Attending school, MVPA in the lower quartile improved. In both subgroups, most strength values and cognitive skills improved, while balance deteriorated in first grade. In the upper quartile, a greater MPA at 6.6 years predicted lower perceptual skills at 7.6 years. A greater SB at 6.6 years predicted higher verbal skills, light and moderate PA and MVPA, and lower verbal skills at 7.6 years after taking into account confounding factors such as the child’s sex, age, awake wear time (AWT), maternal education, and/or child’s sports participation. A vigorous PA at 6.6 years predicted perceptual (in upper quartile) or verbal (in lower quartile) skills at 7.6 years after controlling for similar confounders. No correlation for PF at 6.6 years and cognitive skills at 7.6 years existed; after adjusting for the above-mentioned confounders relative to upper/lower-limb strength, the 4 × 10 m shuttle run results predicted higher perceptual or verbal skills; static balance and cardiorespiratory fitness predicted lower verbal skills. Cardiorespiratory fitness predicted higher perceptual skills after controlling for sex, age, and AWT. Overall, PA and strength were constantly better and SB lower in the upper quartile, yet the lower quartile demonstrated improved MVPA in first grade, and both subgroups increased most components of their strength and cognitive skills in first grade. Higher levels of VPA at kindergarten predicted either better perceptual or verbal skills in first grade after controlling for confounders; the opposite associations were found for other PA levels and cognitive skills in the higher quartile. PF components at kindergarten predicted either superior or inferior cognitive skills in first grade after adjusting for confounders.

## 1. Introduction

The wide range of physical and mental health benefits related to engagement in physical activity (PA) [1,2,3] and high levels of physical fitness (PF) [1,4,5] in childhood and youth are well known, including higher bone mineral density and greater psychological well-being, prevention of overweight/obesity, metabolic syndrome, and psychological ill-being [1,3,4,5]. Some proof for the advantageous influence of PA [2,6,7,8] and PF [4,8,9] on cognitive health or academic success can also be found. On the other hand, high sedentary behavior (SB) has been related to a number of health problems [10]. Regardless of the plethora of evidence that encourages an active and fit lifestyle, there is a worldwide tendency for decreasing levels of PA [11] and PF [12,13]. Preschool years have been described as a critical period with the goal to stay focused on physical activity promotion in order to offer the most invigorating environment for physical and cognitive development during the developmentally critical growth periods [14,15,16]. Still, there are some concerns that transitioning to school will cause life modifications that might be responsible for lower activity levels and longer sedentary time [11,17]. However, research about longitudinal changes in children’s PA, SB, and PF levels and the associations with cognitive performance at early age is limited. Such baseline data along with information about the relationships between health and wellbeing-related variables are essential for future evidence-based PA guidelines and intervention programs [14,18]. Moreover, given that the activity levels between children of the same age might dramatically vary, further elucidation of the longitudinal trajectory patterns and relationships regarding physical and mental health determinants across children displaying either high or low PA levels might be helpful in order to establish more specific and judicious public health strategies related to PA promotion. Hence, in the current study, children were categorized into two groups based on the time spent in moderate-to-vigorous PA (MVPA) levels in the final year of kindergarten. In addition, for informed choices and decisions, it is necessary to understand how potential confounding variables such as the child’s gender and age, total awake wear time (AWT), participation in organized sports, and maternal education affect the relationships between PA and other health-related outcomes.

Therefore, this study had two major goals: (1) to describe the possible differences of PA, SB, PF, and cognitive skills in children subgrouped by their MVPA levels in the final year of kindergarten during the transition from kindergarten to the first grade at school; (2) to investigate the associations between PA and SB at 6.6 years (kindergarten age) with PF and cognitive skills at 7.6 years (school age) using adjustments for several above-mentioned confounders in children classified into two groups according to their MVPA levels in kindergarten.

## 2. Materials and Methods

### 2.1. Study Design and Participants

The current longitudinal study was set in Tartu and nearby counties, Estonia. First, children in their last kindergarten year aged 6–7 years were enrolled from 13 kindergartens, which were randomly selected. The parents of children from 400 families received written information about the study, and agreements were obtained from 284 families. Informed consent was acquired from the parents of all participants. The period for measurements was March to May 2016. About one year later, the same participants were asked to participate in the study again. By that time, children aged 7–8 years studied at first grade in the school. The consent to participate was given by 200 families and the measurements were taken from March to September 2017. The final sample comprised 147 children, of which 51% were boys. Children were categorized into the subgroups of lower and upper quartiles based on their MVPA levels at kindergarten. Approval from the Medical Ethics Committee of the University of Tartu was obtained (reference 254/T-13).

### 2.2. Physical Activity

Children wore the triaxial Actigraph GT3X accelerometer (ActiGraph LLC, Pensacola, FL, USA) on the hip for 7 consecutive days during awake time, with the exception for water-related sports and activities [8,19,20,21]. The accelerometers were set at 15 s epochs. The criteria to include data into the analysis were wear time of at least three days, including one weekend day, with ≥10 h per day of accelerometer use [8,19,20,21,22]. Nonwearing time (at least 20 min of consecutive readings of zero counts and the night-time periods when the unit was removed) was eliminated from the analysis. The cut point for SB was <100 counts per min, and counts of 100 to 1999, 2000 to 3999, and ≥4000 per min were regarded as light PA (LPA), moderate PA (MPA), and vigorous PA (VPA), respectively [8,19,21,23]. The following equation was used to find average daily values of time spent in every PA level: (weekdays × 5 + weekends × 2)/7 [8,21,24]. We found MVPA by summing MPA and VPA [8,19,21]. The sum of SB, LPA, MPA, and VPA was regarded as AWT [8,19,20,21]. LPA, MPA, and VPA were summed up to estimate the total physical activity (TPA).

### 2.3. Body Composition

Body mass and height were measured using medical digital scales (A&D Instruments, Abington, UK) and a portable stadiometer (Seca 213, Hamburg, Germany) to the closest 0.05 kg and 0.1 cm, respectively, with the subject wearing light clothing without shoes [19,20]. To find body mass index (BMI), body mass (kg) was divided by body height squared (m^2^). Age-specific BMI cutoff points were used to determine the overweight [25]. Skinfold thicknesses at two sites (triceps and subscapular) of the body were measured by a Holtain caliper (Crymmych, UK) based on standardized methods [26]. Fat mass (FM; kg) was calculated from skinfold thicknesses using the equations by Slaughter et al. [27], and fat-free mass (FFM; kg) was found by subtracting FM from total body mass [28].

### 2.4. Physical Fitness

Children’s cardiorespiratory fitness (CRF), muscular strength, speed-agility fitness, and static balance were assessed by a standardized PREFIT fitness test battery [29,30,31,32]. CRF was tested with a 20 m shuttle run test, upper-limbs muscular strength was tested with a handgrip strength test, lower-limbs muscular strength was tested with a standing long jump (SLJ), speed-agility fitness was tested with a 4 × 10 m shuttle run test, and static balance was tested with a one-leg stance [31]. The description of the application of these tests has been published earlier [31] and used in our previous studies with children [8,21,23,33,34]. Upper- and lower-limbs muscular strengths were revealed in relation to BMI [35] and FFM [36].

### 2.5. Cognitive Skills

Participants’ cognitive skills were assessed with the Modified Boehm Test of Basic Concepts—Third Edition (Boehm-3) [37], which has been adapted and validated for Estonian children [38,39]. The test is designed to investigate how children at an early age acknowledge the fundamental relational concepts essential for language and cognitive development, and for successful performance at school in all learning spheres. It consists of three separate parts for the assessment of conceptual, verbal, and perceptual skills [37,38,39]. The application of Boehm-3 has been described in our previous studies [8,23]. Briefly, perception, conceptual skills, and verbal abilities were evaluated by separate tests, which were carried out in groups of 3–6 children. The same researcher performed all tests to ensure all the children received similar instructions. Each participant was given a test-book marked with the participant’s code. Conceptual skills were tested to evaluate the understanding of the concepts of adverbs and the place and location of objects. Verbal abilities tested the child’s understanding of the Estonian language. During these tests, the children had to listen to the researcher’s instructions and mark the picture that matched the sentence they were given. The progressive matrix test was used to assess the child’s perception and involved choosing the right picture to complete a regular sequence of pictures. Each right answer scored one point, and the scores were presented for each of the three categories [8,23].

### 2.6. Statistical Analysis

SPSS software (version 20.0; SPSS, Inc., Chicago, IL, USA) was used to analyze the data. Descriptive statistics were presented as means, standard deviations (SD), or frequencies (percentages). The Kolmogorov–Smirnoff test was used for the assessment of normality before the analysis. A paired and independent *t*-test was used to compare means between groups; the chi-square test was applied for testing group differences between categorical variables. In multiple linear regression analysis, the standardized regression coefficient (β) and adjusted coefficients of determination (AdjR^2^) were used to explore the relationships between PA, SB, PF, and cognitive skills. The variance inflation factors in regression analysis were less than five, indicating that collinearity was not a problem. Four regression models were used: the unadjusted model (Model 1); model one adjusted for the age, gender, and AWT of the child at measurements (Model 2); model two adjusted for maternal education at measurements (Model 3); and model three adjusted for the child’s attendance to sports clubs at measurements (Model 4) [8,23]. *p*-value < 0.05 was regarded as statistically significant.

## 3. Results

Children (*n* = 147) were categorized based on their MVPA level in kindergarten into lower and upper quartiles. The characteristics of children at 6.6 and 7.6 years are presented in Table 1. Body composition parameters, the percentage of overweight children, and cognitive test scores did not differ between children in lower quartiles and higher quartiles, neither in kindergarten nor at school. The attendance to sports clubs was greater in children of the upper quartile at 6.6 years than in children of the lower quartile (*p* < 0.05), with no differences between the two quartiles a year later.

In both kindergarten and school settings, children in the upper quartile had less SB and more LPA, MPA, VPA, and MVPA than those of children in the lower quartile (all *p* < 0.05). Children in the upper quartile had greater SLJ test results both at 6.6 and 7.6 years compared to children in the lower quartile (all *p* < 0.05). Children in the upper quartile also had a greater SLS-to-BMI ratio at 6.6 years and they did better in the handgrip strength test at 7.6 years than children in the lower quartile did (all *p* < 0.05). No other differences in PF performance between the two quartiles neither in kindergarten nor at school were observed.

A comparison of children in the lower quartile revealed that body composition parameters were higher at 7.6 years compared to 6.6 years (all *p* < 0.05), except for BMI. Similar changes were present in children of the upper quartile. The percentage of overweight children was not different between the measurement times. Involvement in organized sports decreased at 7.6 years compared to the participation rate at 6.6 years in children of the upper quartile (*p* < 0.05), with no changes in children of the lower quartile. Children in the lower quartile demonstrated at 7.6 years greater SB, MPA, VPA, MVPA, and AWT (all *p* < 0.05) than at 6.6 years. Children in the upper quartile had, by 7.6 years, increased only SB when compared to the age of 6.6 years (*p* < 0.05). There were no other differences among lower or upper quartiles in SB or PA at 7.6 years in comparison to 6.6 years. PF performance improved within one year both among lower and upper quartiles in the following categories—grip strength, grip-to-BMI ratio, SLJ, and SLJ-to-BMI ratio (all *p* < 0.05). In the lower quartile, the grip-to-FFM ratio was higher at 7.6 years than a year earlier (*p* < 0.05). Among both quartiles, the SLJ-to-FFM ratio was higher and the balance performance inferior at 7.6 years compared to the results obtained at 6.6 years (all *p* < 0.05). There was an improvement in perceptual, conceptual, and verbal test scores at 7.6 years compared to data at 6.6 years both among children in the lower and upper quartile (*p* < 0.05).

Table 2 demonstrates the associations of SB and PA at 6.6 years with conceptual skills, verbal abilities, and perception test scores at 7.6 yr in kindergarten MVPA-based children of the lower and upper quartile. Among the upper quartile, a greater SB (*p* < 0.05) at 6.6 years was associated with higher verbal skills at 7.6 years after adjustments for the child’s sex, age, AWT, and maternal education (Model 3), as well as after additional adjustment for sports participation (Model 4). LPA, MPA, MVPA, and TPA (all *p* < 0.05) at 6.6 years were associated with lower verbal test scores at 7.6 years after adjustments for the child’s sex, age, AWT, maternal education (Model 3), and after adding involvement in organized sports further as a covariate (Model 4). On the contrary, greater VPA at 6.6 years was associated with better verbal skills at 7.6 years in both Models 3 and 4 (both *p* < 0.05). In the category of perceptual skills, a negative association between MPA (*p* < 0.05) at 6.6 years and test results at 7.6 years was apparent. No other significant associations between SB and PA levels at 6.6 years with cognitive skills studied a year later were noticed in children of the upper quartile. In children of the lower quartile, VPA at 6.6 years was positively related to perceptual skills at 7.6 years after controlling for the child’s sex, age, AWT, and maternal educational attainment (Model 3), as well as after additional adjustment for attendance to sports clubs (Model 4) (both *p* < 0.05).

Table 3 represents the associations of PF at 6.6 years with conceptual skills, verbal abilities, and perceptual skills at 7.6 years in kindergarten MVPA-based children of lower and upper quartiles. Among children of the upper quartile, a greater number of laps in the 20 m shuttle run test at 6.6 years was related to higher perception test scores (*p* < 0.05) after adjustments for the child’s sex, age, and AWT (Model 2) at 7.6 years, yet lower scores in the verbal reasoning test (*p* = 0.023) after additional adjustment for maternal education (Model 3) a year later. After controlling for confounders such as the child’s sex, age, AWT, and maternal educational attainment (Model 3), greater grip-to-BMI and grip-to-FFM (both *p* < 0.05) ratios at 6.6 years were associated with better perceptual skills at 7.6 years, as well grip-to-BMI ratio to verbal skills one year later. A further adjustment with sports participation (Model 4) resulted in similar positive associations between grip-to-FFM ratio (*p* < 0.05) at 6.6 years and perceptual skills at 7.6 years. After adjusting for confounders such as the child’s sex, age, AWT, and mother’s education (Model 3), a greater SLJ-to-BMI ratio at 6.6 years was associated with better perceptual and verbal skills (both *p* < 0.05) at 7.6 years. A greater SLJ-to-FFM ratio at 6.6 years was related to better perceptual skills after adjusting for confounders such as the child’s sex, age, AWT, and mother’s education (Model 3), and further adding the engagement in organized sports (Model 4) (both *p* < 0.05). A better performance in the 4 × 10 m shuttle run test at 6.6 years was related to a higher score on the verbal ability test after adjusting for confounders such as the child’s sex, age, AWT, and maternal education (Model 3), and further adding the attendance to sports clubs (Model 4) (both *p* < 0.05). Static balance test results at 6.6 years were negatively associated with verbal skills (*p* < 0.05) at 7.6 years after confounding through the child’s sex, age, AWT, and mother’s education adjustment (Model 3). We did not find any other associations between PF at 6.6 years and cognitive abilities at 7.6 years neither in unadjusted nor adjusted models in children of the upper quartile. In children belonging to the lower quartile, positive associations between the number of laps in the 20 m shuttle run test at 6.6 years with verbal test scores at 7.6 years after controlling for confounders including the child’s sex, age, AWT, maternal education (Model 3), and their involvement in organized sports (Model 4) (both *p* < 0.05) were detected. Better performance in the 4 × 10 m shuttle run test at 6.6 years was related to a lower score on the verbal ability test after adjusting for confounders such as the child’s sex, age, and AWT (Model 2) (*p* < 0.05). No other associations between PF at 6.6 years and cognitive abilities at 7.6 years were observed neither in unadjusted nor adjusted models in children of the lower quartile.

## 4. Discussion

The current study aimed to explore longitudinally the possible differences and associations of PA, SB, PF, and cognitive abilities in children of two extremes with regard to their physical activity levels at preschool age. Children were categorized into lower and upper quartiles based on their MVPA level at 6.6 years at preschool age, and their activity, fitness, and cognitive competencies-related development trajectories into the first grade at school were assessed.

### 4.1. Description of PA, SB, PF, and Cognitive Skills in the Transition from Preschool to School in Children Subgrouped by Their MVPA at 6.6 Years

First, the present study demonstrates that kindergarten MVPA-based children of the upper quartile had lower SB and higher LPA, MPA, VPA, and MVPA both at 6.6 and 7.6 years in comparison to the children in the lower quartile. In addition, children in the upper quartile had greater SLJ test results both at 6.6 and 7.6 years compared to children in the lower quartile. Children in the upper quartile also had a greater SLS-to-BMI ratio at 6.6 years and upper-limbs muscular strength at 7.6 years than children in the lower quartile.

Although the benefits of PA are well-known, a decline in total PA has been observed after elementary school years and the transition to young adulthood [40]. The transition from kindergarten to school is also a lifechanging event for children and a decline in PA accompanied with increased sedentary time has been reported previously [11]. Few data exist about PA change in youth [41] and adulthood [42], especially according to their PA level in preschool years. However, the results of our study demonstrated the stability of PA patterns in both subgroups of children. The children of our study who were more active in kindergarten showed higher levels of PA also after entering school. Our children of the upper quartile exceeded their peers of the lower subgroup in time engaged in MVPA at the school, as well as in kindergarten, exceeding firmly the recommended minimum of 60 min of daily MVPA across the week [43]. However, children in the lower quartile significantly improved (on average by 15 min) their MVPA levels upon entering school. The Estonian school setting provides several options for PA. First, in Estonia, physical education is a compulsory subject for basic and upper secondary schools [44,45], and in grades 1–3, it is mandatory to provide 8 weekly *hours* of physical education classes [44], for instance, 2, 3, and 3 h, respectively. Second, in 2016, the Estonian nationwide comprehensive physical activity program Schools in Motion (the SiM) was initiated, and by the year of 2017, four schools that participated in the current study had joined The SiM program. The program focuses on providing various ideas and options for physical activities during breaks and after school, as well encourages *active breaks* as short bursts of physical activity in the classroom and more outdoor activities during breaks, having basic school (grades 1–9, ages 7–16 years) children as the main target group. The idea is not simply to increase activity levels at school, but to form healthy lifestyle habits for the rest of life. By 2020, as many as 110 schools all over Estonia have joined the SiM program [46]. In addition, children in the Tartu region are quite independent while attending school, and the distance to reach school is often quite short, so walking, biking, and common transport are frequently used. There are also many options for different organized sports programs either in school or sports centers all over the city; for instance, soccer, track and field, and acrobatics were the most popular training activities among children in current study.

The time spent sedentary increased in both subgroups, which can be explained by the school curriculum. The stability of physical activity patterns among the participants of our study has also been expressed in the dynamics of physical fitness indices of both subgroups. The children of the upper quartile showed a higher enhancement in cardiorespiratory fitness and, at the same time, they participated more in organized sports. Sports club attendance has also been demonstrated to have a stronger relationship with physical fitness compared to unstructured PA. Higher fitness may further contribute to higher interest in PA and enhance the chances to continue with an active lifestyle [47].

### 4.2. The Associations of PA and SB with PF in the Transition from Preschool to School in Children Subgrouped by Their MVPA at 6.6 Years

The second goal of the present study was to shed light on the relationships between PF in children of the MVPA-based upper and lower quartile at the age of 6.6 years with their cognitive abilities at 7.6 years. In general, most associations among upper and lower quartiles confirmed a link between the higher physical capabilities of preschoolers and cognitive skills either among verbal or perceptual domains at school in adjusted analyses. The previous results, obtained from the same children involved in the present study without dividing children into subgroups, substantiated the positive impact of CRF, relative explosive strength of the lower limbs, and static balance at 6.6 years on conceptual and perceptual skills a year later [8]. Academic benefits of clustered PF and CRF have been reported by other studies in children, yet the assumptions have mostly been made based on cross-sectional data [9]. Besides many positive outcomes with regard to the effects of PF on cognitive abilities seen in present study, some results were unexpected. We found a negative impact of speed-agility skills at 6.6 years on verbal abilities one year later in children of the lower quartile after adjusting for gender, age, and AWT. However, after confounding with more covariates, the associations were not more significant. Surprisingly, negative relations appeared between CRF and static balance at 6.6 years with verbal skills at 7.6 years among the upper quartile, after adding into analysis the maternal education as a characteristic of SES. In health research, no single gold standard marker of SES exists [48,49], and parental education is a commonly used indicator of overall SES in research focused on the sedentary behavior, physical activity, and fitness in children [6,18,23,50]. Yet, some have preferred to use solely parent-reported household income as a determinant of SES [51], whereas some have added marital status [6]. Moreover, the impact of SES on SB has been shown to vary with respect to differences in income levels of countries, and by specific domains of SB [52]; furthermore, a recent umbrella review found no association between PA and parental SES for children and adolescents, keeping in mind that the conclusion was based on limited evidence [53]. Therefore, our findings that children born to mothers with higher educational attainment have lower verbal test scores at 7.6 years in relation to some domains of PA or fitness at 6.6 years should be interpreted with reasonable caution. Otherwise, similar mechanisms are related to possible reductions in mother–kid interactions because of occupational duties and complexed sports participation-related logistics, allowing time for a child to collect activity minutes, thus improving their cardiovascular health and muscle fitness. We further noticed that the above-described negative effect of a higher number of laps in the 20 m shuttle run test and static balance performance at 6.6 years on verbal skills at 7.6 years was reversed by additional confounding with participation in sports trainings, and the associations were no longer significant. Confounding with more variables caused the disappearance of some positive effects of PF measures, and a still-greater grip-to-FFM ratio, grip-to-BMI ratio, SLJ-to-FFM ratio, and better performance on the 4 × 10 m shuttle run test at 6.6 years remained significantly related to higher cognitive test scores after adding engagement in sports clubs into analysis.

### 4.3. The Associations of PA and SB with Cognitive Skills in the Transition from Preschool to School in Children Subgrouped by Their MVPA at 6.6 Years

Adequate *cognitive* performance is a prerequisite for children’s effective functioning in a school environment [50]. Physical exercise is considered to have an important role in child cognitive development [2,54]. Voluntary movement, as a result of signals from the brain travelling and reaching the muscles, induces displacements of body parts. Information from the periphery helps to form a feedback control loop [55]. That is why certain specific neuronal adaptations take place, including changes that support memorial and learning benefits [56], as confirmed by findings where less SB and more MVPA were linked to greater gray matter volume in the right hippocampus in overweight preadolescent children [24]. Still, while cognitive benefits of acute and chronic exercise in adults have been widely reported [57,58,59], there is limited and controversial information about the impact of PA on cognitive performance in children [2,6,7,8], and several studies could not find data to support the role of physical exercises with regard to cognitive development [23,50,60,61].

The present study focused mainly on investigating the associations between the PA and SB of children of the MVPA-based upper and lower quartile at the age of 6.6 years with their cognitive abilities at the age of 7.6 years. The general characteristics of children showed that perceptual, conceptual, and verbal test scores were higher at 7.6 years than at 6.6 years in children of both lower and upper quartiles. Yet, cognitive test scores were not different between children in the lower and higher quartiles, neither when children were 6.6 years nor 7.6 years old. The time that children in the upper quartile spent on vigorous activities was twice (at 7.6 years) to thrice (6.6 years) as much as in children of the lower quartile. In respect of verbal abilities, linear regression analysis using confounding with the child’s sex, age, AWT, and maternal education and/or, additionally, participation in organized sports stresses out the importance of high levels of VPA in the preschool period in children of the upper quartile, linking it to better verbal reasoning test scores measured one year later in the first grade at school. In addition, the importance of vigorous activities was confirmed in children of the lower quartile, where the addition of possible confounders at 6.6 years was associated with better perceptual competence at 7.6 years. Greater levels of VPA have also been related to higher grades in sixth-grade students, especially among those following the Healthy People 2010 recommendations for at least 20 min VPA per day at least 3 days per week. At the same time, the amount of MPA was not associated with higher grades [62]. In addition, research has, in turn, connected higher perceptual abilities with better whole academic performance among 4–6-graders [63], indicating the importance of current findings about the relations between VPA and perceptual skills in the context of academic success. Practicing vigorous activities has been shown to be more advantageous than MPA, as well as in terms of general health outcomes for youths [64]. Our previous study among the same total sample that participated in current study demonstrated that greater levels of physical activity of all intensities (LPA, MPA, VPA, and MVPA) at 6.6 years were associated with higher test results, and SB with a lower score in the examination of conceptual skills at 7.6 years, after controlling for possible confounders. To summarize, we found that practicing PA at recommended levels has a positive impact on conceptual skills, and it does not influence verbal or perceptual skills [8]. The current results demonstrate that, among the very active preschoolers, VPA levels positively affect their later verbal skills, encouraging activities such as running, hopping, rope jumping, active biking, ice-hockey, playing ballgames, and other active games. Still, data from the present study reveal that the associations between the PA of preschoolers in high-activity categories with their later cognitive abilities at school are not so unambiguous considering other physical activity intensities but VPA. Adding to covariates such as the child’s sex, age, AWT, and, additionally, maternal education, LPA, MPA, MVPA, or TPA at 6.6 years was associated with lower verbal skills at 7.6 years in children of the upper quartile. On the contrary, similar adjustments for SB at 6.6 years were linked to better scores in the verbal test a year later.

### 4.4. The Role of Maternal Education in Determining the Relations between PA and Cognitive Skills in the Transition from Preschool to School in Children Subgrouped by Their MVPA at 6.6 Years

It is well known that socioeconomic status (SES) is an important factor influencing cognitive, social, and brain development [65]. It has been reported that the mother’s educational status as an important indicator of SES is the most strongly associated with children’s cognitive development and well-being [66], and language is one of the cognitive spheres most touched by SES [67]. Despite that, we found that the synergy between all other PA intensity levels but VPA with increasing mother’s educational attainment attenuated verbal performance in children of higher quartiles, up to an extent, where adding the participation in sports trainings as a covariate did not have a protective effect. We theorize that increased rates of employment among mothers with higher education, and perhaps continuing working at home after official workdays, mean the struggle to balance work and family commitments and less nonwork time available to devote to children, which may have some negative impacts on the children’s verbal skills. Craig et al. [68] studied the influence of parental education on the time spent with children in 1992 and 2006, and they concluded that education had no effect on total time with children or on recreation and social time with children in neither time period. Mothers who worked part-time spent significantly more time with their children, but not significantly more in recreational and social activities with their children. Leisure time activity of the age group of this study depends very much on the contribution of parents, and previous studies have shown a higher attendance of sports clubs among the children of highly educated families [23]. In the current study, 95.8% of children in the upper quartile participated in sports trainings, while much fewer (62.5%) children in the lower quartile were involved in organized sports activities. One factor reducing mother–kid social interactions and thus influencing their verbal development could also be the time lost on transportation logistics and often waiting for the child to complete the workout on spot. In addition, as there is a certain number of hours per day, if children spend more time on physical activities, there might be less possibilities to develop their verbal skills, if the environment where the activities take place does not encourage speech and language development. Hence, there might be time lost for adult–child two-sided conversations and reading with the child or storytelling for healthy language maturation [69]. We did notice that a longer sedentary time at 6.6 years was associated with better verbal skills one year later after adjusting for potential confounders in children belonging to the upper quartile. This, in turn, underpins the importance of finding a proper balance of activities in the everyday life of a preschooler and their family.

## 5. Strengths and Limitations

The limitations of the current study include a relatively small sample size; therefore, the possibility exists to be underpowered to detect associations between some of the PA or PF components with cognitive skills. In addition, body composition was measured indirectly using skinfolds. Although skinfold thickness used in the current study is described as a more sensitive marker than BMI in calculating body fat, DEXA is still considered to be the gold standard for measuring body fat composition in children. These limitations are balanced by several strengths. We used accelerometers to objectively measure PA, and the assessment of physical fitness was performed by a standardized test battery and controlled for several confounders when analyzing associations between PA, physical fitness indices, and body composition variables with components of cognitive skills.

## 6. Conclusions

The novelty of the present study was to illustrate the complexity of longitudinal associations between objectively measured PA, SB, and PF with cognitive performance among children of two extremes, who were grouped according to their MVPA levels at preschool age. The physical activity patterns were stable, although the children of both subgroups showed increased results in physical fitness in the first grade at school. A higher level of VPA at kindergarten was positively associated with cognitive skills a year later both in upper and lower quartiles after controlling for confounders.

## Figures and Tables

**Table 1 ijerph-18-07576-t001:** Descriptive statistics of study participants at 6.6 years (final year of kindergarten) and at 7.6 years (1st grade at school).

	Kindergarten		School	
	Lower Quartile (*n* = 24)	Upper Quartile (*n* = 24)	Lower Quartile (*n* = 27)	Upper Quartile (*n* = 28)
**Variable**				
Age (yr)	6.57 ± 0.6	6.48 ± 0.51	7.67 ± 0.48 *	7.48 ± 0.5 *
Height (cm)	124 ± 6.15	127 ± 6.4	131 ± 6.9 *	134 ± 6.4 *
Weight (kg)	24.8 ± 5.1	26.24 ± 4.55	28.3 ± 6.75 *	29.9 ± 5.8 *
BMI (kg/m^2^)	16.0 ± 2.33	16.22 ± 1.66	16.3 ± 2.95	16.6 ± 2.11
Overweight (%)	25%	20.8%	11.1%	10.7%
Participating in organized sport (%)	62.5%	95.8% #	63%	75%
**SB and PA**				
AWT (min/day)	803 ± 143	805 ± 82	862 ± 107 *	837 ± 94
SB (min/day)	460 ± 141	381 ± 72.6 #	506 ± 103 *	425 ± 80 *#
Light PA (min/day)	298 ± 55	325 ± 28 #	296 ± 42	320 ± 39 #
Moderate PA (min/day)	33.3 ± 4.9	64 ± 12 #	41 ± 11.4 *	61 ± 15.1 #
Vigorous PA (min/day)	11.1 ± 3.8	34 ± 11.5 #	17.76 ±10.87 *	34 ± 15.5 #
MVPA (min/day)	44.3 ± 6.8	99 ± 19 #	59 ± 20.4 *	95 ± 28.5 #
**PF tests**				
20 m shuttle run (laps)	17.1 ± 9.47	20.0 ± 11.5	20 ± 13.7	25.6 ± 15.1
Grip strength (kg)	10.3 ± 1.93	11.5 ± 2.4	12.9 ± 2.41 *	13.7 ± 2.6 *#
Grip-to-BMI ratio	0.65 ± 0.14	0.71 ± 0.15	0.8 ± 0.16 *	0.83 ± 1.14 *
Grip-to-FFM ratio	0.54 ± 0.08	0.55 ± 0.08	0.58 ± 0.79 *	0.55 ± 0.06
SLJ (cm)	114 ± 18.9	129 ± 16.4 #	124 ± 23.5 *	139.1 ± 19.6 *#
SLJ-to-BMI ratio	7.16 ± 1.71	8.2 ± 1.27 #	7.91 ± 2.32 *	8.6 ± 1.66 *
SLJ-to-FFM ratio	5.96 ± 1.31	6.43 ± 1.07	5.6 ± 1.5 *	5.81 ± 1.09 *
4 × 10 m shuttle run (s) ^a^	15.6 ± 1.6	14.9 ± 1.11	15.02 ± 1.53	14.5 ± 0.98
One-leg stance (balance) (s)	21 ± 9.87	22.1 ± 8.6	12.5 ± 9.4 *	13.1 ± 9.04 *
**Modified Boehm-3 test**				
Progressive matrix (max score 10)	6.26 ± 2.68	6.60 ± 2.57	7.70 ± 2.16 *	7.83 ± 1.9 *
Conceptual skills (max score 17)	13.3 ± 2.72	13.8 ± 1.9	14.9 ± 1.41 *	14.6 ± 1.7 *
Verbal abilities (max score 9)	5.58 ± 1.40	5.63 ± 1.54	6.5 ± 1.1 *	6.8 ± 1.2 *

Data are given as mean ± SD. BMI: body mass index; FFM: fat-free mass; SB: sedentary behavior; PA: physical activity; AWT: total awake wear time; MVPA: moderate-to-vigorous physical activity; TPA: total physical activity; PF: physical fitness; SLJ: lower-limbs muscular strength. ^a^ The lower the score (in s), the higher the performance. * *p* < 0.05 lower quartile in kindergarten vs. lower quartile at school; higher quartile in kindergarten vs. higher quartile at school. # *p* < 0.05 lower quartile in kindergarten vs. higher quartile in kindergarten; lower quartile at school vs. upper quartile at school.

**Table 2 ijerph-18-07576-t002:** Associations of sedentary behavior and physical activity intensities at 6.6 years (final year of kindergarten) with cognitive abilities at 7.6 years (1st grade at school) in kindergarten MVPA-based children of lower (25%) and upper (75%) quartiles.

	Cognitive Abilities at 7.6 Years											
	Lower Quartile						Upper Quartile					
	Conceptual Skills		Verbal Abilities		Perception		Conceptual Skills		Verbal Abilities		Perception	
**SB and PA at 6.6 years**	AdjR^2^	β	AdjR^2^	β	AdjR^2^	β	AdjR^2^	β	AdjR^2^	β	AdjR^2^	β
SB												
Model 1	−0.042	−0.057	−0.040	−0.075	−0.037	0.088	−0.044	0.043	−0.045	0.016	−0.043	−0.048
Model 2	−0.268	−0.228	−0.227	−0.065	0.110	−1.016	0.300	−0.953	0.165	0.562	0.206	1.566
Model 3	−0.768	−0.009	−0.026	−0.170	0.131	−1.075	0.475	−3.363	0.932	2.281 *	0.419	3.741
Model 4	0.395	2.849	0.188	4.092	0.948	0.265	0.729	−4.555	0.918	2.371 *	0.337	3.255
LPA												
Model 1	−0.043	0.047	−0.039	−0.078	−0.044	0.041	−0.038	−0.084	−0.041	0.069	−0.045	0.002
Model 2	−0.270	0.051	−0.227	−0.015	0.085	0.254	0.319	0.589	0.161	0.303	0.043	−0.573
Model 3	−0.767	−0.024	−0.031	0.012	0.058	0.271	0.404	−0.679	0.864	−0.281 *	0.235	−0.457
Model 4	0.603	−0.894	0.522	−1.262	0.955	−0.100	0.401	−0.517	0.831	−0.278 *	0.158	−0.313
MPA												
Model 1	−0.044	0.037	−0.044	0.041	−0.044	0.041	−0.036	−0.093	0.019	−0.249	0.134	−0.415 *
Model 2	−0.241	0.154	−0.159	0.227	0.198	0.376	0.292	0.288	0.241	−0.314	0.303	−0.573
Model 3	−0.712	0.171	0.042	0.196	0.469	0.548	0.458	0.388	0.918	−0.252 *	0.382	−0.416
Model 4	−0.392	0.586	−1.731	0.619	0.980	0.215	0.681	0.514	0.898	−0.258 *	0.304	−0.358
VPA												
Model 1	−0.035	−0.101	−0.044	0.040	−0.045	0.002	−0.015	−0.170	−0.019	−0.160	−0.031	−0.116
Model 2	−0.273	−0.010	−0.226	0.038	0.066	0.218	0.205	−0.107	0.197	−0.323	−0.072	−0.038
Model 3	−0.753	0.121	0.289	0.555	0.651	0.837 *	0.455	0.548	0.895	0.298 *	0.318	0.490
Model 4	0.792	1.171	0.404	1.508	0.998	0.234 *	0.422	0.406	0.871	0.318 *	0.217	0.375
MVPA												
Model 1	−0.045	−0.015	−0.045	0.013	−0.045	0.030	−0.023	−0.147	0.014	−0.238	0.059	−0.316
Model 2	−0.257	0.108	−0.186	0.178	0.174	0.355	0.292	0.463	0.257	−0.385	0.116	−0.467
Model 3	−0.725	0.155	0.100	0.267	0.566	0.592	0.635	0.734	0.850	−0.102 *	0.198	−0.169
Model 4	0.034	0.876	−1.042	1.039	0.994	0.240	0.750	0.727	0.815	−0.103 *	0.153	−0.175
TPA												
Model 1	−0.044	0.039	−0.041	−0.068	−0.044	0.043	−0.025	−0.139	−0.036	−0.093	−0.011	−0.181
Model 2	−0.268	0.064	−0.227	0.018	0.110	0.287	−0.300	0.437	0.165	−0.258	0.206	−0.718
Model 3	−0.768	0.003	−0.026	0.052	0.131	0.325	0.475	1.403	0.932	−0.952 *	0.419	−1.560
Model 4	0.395	−0.906	0.188	−1.302	0.948	−0.084	0.729	1.900	0.918	−0.989 *	0.337	−1.338

Model 1: unadjusted; Model 2: Model 1 + gender, age, and total awake wear time at measurements; Model 3: Model 2 + maternal educational attainment at measurements; Model 4: Model 3 + sports clubs attendance at measurements. AdjR^2^: adjusted coefficient of determination; β: standardized regression coefficient; SB: sedentary behavior; PA: physical activity; LPA: light physical activity; MPA: moderate physical activity; VPA: vigorous physical activity; MVPA: moderate-to-vigorous physical activity; TPA: total physical activity. * *p* < 0.05.

**Table 3 ijerph-18-07576-t003:** Associations of physical fitness at 6.6 years (final year of kindergarten) with cognitive abilities at 7.6 years (1st grade at school) in kindergarten MVPA-based children of lower (25%) and upper (75%) quartiles.

	Cognitive Abilities at 7.6 Years											
	Lower Quartile						Upper Quartile					
	Conceptual Skills		Verbal Abilities		Perception		Conceptual Skills		Verbal Abilities		Perception	
**Physical fitness at 6.6 years**	AdjR^2^	β	AdjR^2^	β	AdjR^2^	β	AdjR^2^	β	AdjR^2^	β	AdjR^2^	β
Cardiorespiratory fitness												
Model 1	0.047	0.307	−0.010	0.200	0.112	0.395	−0.006	0.224	−0.059	0.008	−0.047	0.104
Model 2	−0.137	0.585	−0.126	0.065	0.362	0.645	0.292	0.363	0.011	0.207	0.536	0.375 *
Model 3	0.040	0.940	0.550	−0.096	0.222	0.001	0.668	0.651	0.844	−0.036 *	0.661	0.048
Model 4	−0.076	0.872	0.615	−0.178	0.082	0.046	0.596	0.575	0.794	−0.022	0.606	0.358
Grip-to-BMI ratio												
Model 1	−0.038	−0.117	−0.016	−0.186	−0.033	0.138	−0.005	−0.225	−0.044	0.119	−0.043	−0.121
Model 2	−0.394	−0.059	−0.108	−0.196	0.189	0.511	0.125	−0.095	−0.004	−0.189	0.389	0.188
Model 3	−0.483	−0.196	0.773	0.567 *	0.261	0.234	0.178	0.311	0.857	0.107 *	0.842	0.469 *
Model 4	−0.591	−0.263	0.831	0.523 *	0.143	0.268	0.160	0.195	0.819	0.130	0.840	0.521
Grip-to-FFM ratio												
Model 1	−0.040	−0.145	0.032	−0.298	0.025	0.287	−0.037	−0.155	0.014	−0.269	−0.050	−0.109
Model 2	−0.313	0.176	−0.188	−0.226	0.293	0.166	0.394	−0.186	0.047	−0.434	0.266	0.071
Model 3	0.927	−0.019	−0.498	0.979	0.713	−0.141	0.194	0.142	0.801	0.073	0.931	0.585 *
Model 4	0.910	0.257	−0.002	0.921	0.976	−0.999	0.194	0.142	0.801	0.073	0.931	0.585 *
SLJ-to-BMI ratio												
Model 1	−0.026	0.158	−0.042	−0.101	0.064	0.333	−0.032	−0.170	0.039	−0.150	−0.058	0.063
Model 2	−0.394	0.049	0.046	−0.431	0.408	0.619	0.119	−0.059	0.062	−0.273	0.356	0.085
Model 3	−0.391	−0.373	0.568	−0.165	0.467	0.536	0.065	−0.064	0.874	0.126 *	0.823	0.360 *
Model 4	−0.538	−0.328	0.131	−0.126	0.364	0.518	0.177	0.175	0.832	0.126	0.777	0.379
SLJ-to-FFM ratio												
Model 1	0.033	0.300	−0.050	−0.109	0.134	0.430	0.099	0.394	0.060	−0.345	0.020	0.285
Model 2	−0.243	0.248	−0.121	−0.261	0.443	0.339	0.353	−0.054	0.118	−0.541	0.260	0.000
Model 3	0.927	0.028	0.281	−0.427	0.789	0.257	0.192	0.162	0.806	0.106	0.885	0.658 *
Model 4	0.902	0.103	0.983	−0.809	0.629	0.185	0.192	0.162	0.806	0.106	0.885	0.658 *
Speed-agility fitness ^a^												
Model 1	0.031	0.283	0.050	0.312	0.048	−0.309	0.116	0.406	0.056	−0.051	−0.054	−0.067
Model 2	−0.166	0.532	0.477	0.862 *	0.241	−0.487	0.126	0.118	0.030	−0.385	0.427	−0.300
Model 3	0.214	1.058	0.601	0.296	0.553	−0.715	0.091	0.217	0.882	−0.233 *	0.604	−0.227
Model 4	0.204	1.675	−0.597	−0.146	0.646	−1.362	0.681	0.994	0.910	−0.447 *	0.529	−0.423
Balance												
Model 1	−0.038	0.118	−0.044	0.090	0.106	0.388	−0.052	0.080	−0.056	−0.052	−0.025	0.178
Model 2	−0.392	0.065	−0.128	0.024	0.188	0.395	0.198	−0.285	0.002	−0.219	0.356	−0.101
Model 3	−0.448	−0.262	0.568	−0.162	0.329	0.345	0.069	0.102	0.850	−0.076 *	0.567	0.013
Model 4	−0.331	−0.600	0.753	−0.427	0.365	0.565	0.292	−0.490	0.800	−0.092	0.429	0.080

Model 1: unadjusted; Model 2: Model 1 + gender, age, total awake wear time at measurements; Model 3: Model 2 + maternal educational attainment at measurements; Model 4: Model 3 + sports clubs attendance at measurements. AdjR^2^: adjusted coefficient of determination; β: standardized regression coefficient. ^a^ The lower the score (in s), the higher the performance. BMI: body mass index; FFM: fat-free mass; SLJ: lower-limbs muscular strength * *p* < 0.05.

## Data Availability

The data presented in this study are available on request from the corresponding author. The data are not publicly available due to restrictions e.g., their containing information that could compromise the privacy of research participants.

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
