# Peer review of "Physical Activity, Fitness, and Cognitive Performance of Estonian First-Grade Schoolchildren According Their MVPA Level in Kindergarten: A Longitudinal Study"

_ijerph, 2021, doi:10.3390/ijerph18147576_

Round 1
Reviewer 1 Report
The paper is well written in terms of better understanding between school children physical activity and cognitive development.
The research design is acceptable by using longitudinal cohort approach.
line 61: data collected except water-related sport. However, the second data collection period was from March to September and summer sport activities were involved. A description of the Estonian children's physical activity types can provide more details to the readers.
The four statistical analysis models and the results are well presented.
For discussion part, the paragraphs are too long and it should break down into shorter paragraph on each focusing issue. For example, the content of Estonian school physical education programs, after-school sport, inter-school sport competition, home-based physical activity, etc.
line 262 to 343: paragraph is too long.
line 292: how is the present finding related to the literature? more explanation should be included.
line 331: provide some examples in so-called 'sports trainings', e.g. ball games, gymnastics, free-play, etc.
line 340 to 344: provide some practical suggestion
Overall speaking, the study is interesting to understand the Estonian school children behavior between physical activity and cognitive development.
Author Response
"Please see the attachment"

Reviewer 2 Report
An interesting study in which the authors attempt to track the trajectories of physical activity and cognitive skills in the transition from kindergarten to first grade. I have a number of serious concerns about the introduction, methods and discussion which need addressing.
General comments:
Introduction.
Generally lacking in information. You basically just say PA is good and SB is bad. Could do with some elaboration about the specific benefits for children. Introduction just peters out and doesn't really end with an aim.
Methods.
You need to make it much clearer that this is a longitudinal study. From both the methods and abstract, it actually reads like a an observational study of two different cohorts. It is only when you get to the discussion that things really become clearer, when you talk about trajectory into first grade.
Discussion.
Excessively long and rambling paragraphs, with much speculation. Needs to be much more concise and focused.
Specific comments:
Abstract, lines 18 and 20. What were the confounders?
Introduction, line 28. Guessing you mean data rather than date?
Introduction, lines 32-37. Long sentence. Could do with breaking up into two sentences, to give better flow.
Methods, line 74. Could prefer you briefly describe the methods for determining height and weight here, rather than just referring to previous papers.
Methods, line 95. For those people who are unfamiliar with the Boehm-3, a little further description here would be useful. Just a sentence or two.
Methods, line 110. Should the adjustments for models 3 and 4 not be mentioned briefly earlier? Maybe the introduction? Seems a bit of a shock just introducing these at the end of the methods.
Discussion, line 245. Got a reference for this? Seems like quite an anecdotal statement.
Discussion, lines 223-261. Massive paragraph. Consider breaking up. For example, one paragraph talking about school and another about organised sports. Same with the next paragraph, which is even longer (lines 262-343).
Author Response
"Please see the attachment"

Round 2
Reviewer 2 Report
I thank the authors for addressing my comments. Their manuscript is much improved. I still have several minor concerns before I can accept.
Line 15. I find from line 15 to the end of the abstract quite difficult to follow. At times, it seems like there are grammatical errors which mean that sentences run into each other. The abstract is the first thing people will read, so needs to be well-written and easy to understand.
Lines 35-39. This sentence is rather long-winded and difficult to understand. I would suggest splitting it in two.
Lines 143-148. I should spotted this before, but your descriptions of the models are a little confusing. Your unadjusted model is named model 1, but when talking about model 2 you also call it model 1. The same issue persists for the other two models.
Line 344-427. Whilst you have broken the discussion up into subheadings, some paragraphs are still incredibly long. This one I have highlighted is nearly 100 lines long. It needs to be split up into several shorter paragraphs to make it easier to read.
